Intuitive physics and intuitive psychology (“theory of mind”) in offspring of mothers with psychoses

Maróthi Rebeka 1
Kéri Szabolcs 1 2 3 keri.szabolcs.gyula@med.u-szeged.hu szkeri2000@yahoo.com
1 National Psychiatry Center , Budapest , Hungary
2 Department of Physiology, Faculty of Medicine, University of Szeged , Szeged , Hungary
3 Department of Cognitive Science, Budapest University of Technology and Economics , Budapest , Hungary
Cloninger C. Robert
Electronic publication date: 2014 Mar 27
Publication date: 2014
Volume: 2
Electronic Location ID: e330
Received 2014 Feb 19; Accepted 2014 Mar 13
Copyright: © 2014 Marothi et al.
Copyright year: 2014
Copyright holder: Marothi et al.
License: This is an open access article distributed under the terms of the Creative Commons Attribution License, which permits unrestricted use, distribution, reproduction and adaptation in any medium and for any purpose provided that it is properly attributed. For attribution, the original author(s), title, publication source (PeerJ) and either DOI or URL of the article must be cited.
License URL: https://creativecommons.org/licenses/by/4.0/

Keywords: Social cognition, Folk psychology, Schizophrenia, Theory of mind, Bipolar disorder

Funding: National Brain Research Project The study was supported by the National Brain Research Project. The funders had no role in study design, data collection and analysis, decision to publish, or preparation of the manuscript.

==============================
Offspring of individuals with psychoses sometimes display an abnormal development of cognition, language, motor performance, social adaptation, and emotional functions. The aim of this study was to investigate the ability of children of mothers with schizophrenia (n = 28) and bipolar disorder (n = 23) to understand mental states of others using the Eyes Test (folk psychology or “theory of mind”) and physical causal interactions of inanimate objects (folk physics). Compared with healthy controls (n = 29), the children of mothers with schizophrenia displayed significantly impaired performances on the Eyes Test but not on the folk physics test when corrected for IQ. The children of mothers with bipolar disorder did not differ from the controls. The folk physics test showed a significant covariance with IQ, whereas the Eyes Test did not exhibit such covariance. These results suggest that the attribution of mental states, but not the interpretation of causal interaction of objects, is impaired in offspring of individuals with schizophrenia, which may contribute to social dysfunctions.

Introduction

Several studies provided evidence that offspring of individuals with schizophrenia display an abnormal development of cognition, language, motor performance, social adaptation, and emotional functions (Niemi et al., 2003). Children who later develop bipolar disorder also show developmental and behavioral abnormalities, but these are less severe than that seen in children who later develop schizophrenia (Murray et al., 2004). In spite of extensive research in the field, the basic mechanisms of social and cognitive impairments in children of individuals with psychoses are poorly understood.

To elucidate this issue, we selected the evolutionary framework of folk psychology and folk physics, which is based on the classification of objects in our environment as agents and non-agents (Dennett, 1987; Sperber, Premack & Premack, 1995; Csibra et al., 1999; Baron-Cohen et al., 2001). Agents have intentionality and can move by self-propulsion. Non-agents have no intentionality and their movements are initiated by another object. From infancy, humans use folk or intuitive psychology to deduce the mental causes of agents’ actions and use folk or intuitive physics to deduce the physical causes of non-agents’ movements. The term “theory of mind” is closely related to folk psychology and refers to the ability to infer one’s own and other persons’ mental states. Evidence suggests that “theory of mind” is impaired in patients with schizophrenia and may contribute to language and social dysfunctions (Frith, 2004; Brune, 2005). However, “theory of mind” is also impaired in patients with bipolar disorder (Kerr, Dunbar & Bentall, 2003; Bora et al., 2005) and depression (Lee et al., 2005), which suggests that it is not specific to schizophrenia.

The aim of this study was to investigate folk psychology (“theory of mind”) and folk physics in children of mothers with schizophrenia and psychotic bipolar disorder using the methods of Baron-Cohen et al. (2001). We addressed the following questions: (i) Are folk psychology and folk physics equally affected in the children of mothers with schizophrenia and bipolar disorder? (ii) Are the offspring of individuals with schizophrenia and bipolar disorder impaired relative to controls with negative family history for psychotic disorders? Our hypothesis was that children of mothers with schizophrenia would display a significant and generalized deficit in both folk psychology and folk physics tests, whereas the children of mothers with bipolar disorder would be much less severely affected (Murray et al., 2004).

Materials and Methods

Participants

Mothers with schizophrenia and with type I bipolar disorder with psychotic features were recruited at the National Psychiatric Center, Budapest, and University of Szeged, Szeged, Hungary. Parents of control participants were university and hospital employees and their relatives or acquaintances. Diagnoses were based on the DSM-IV criteria (American Psychiatric Association, 1994). All mothers received the MINI International Neuropsychiatric Interview Plus (Sheehan et al., 1998) and their full medical records were available. Mothers with schizophrenia and type I bipolar disorder did not differ in illness duration (schizophrenia: 12.8 years, SD = 5.6; bipolar disorder: 14.2, SD = 8.2) and in the number of acute hospitalizations (schizophrenia: 6.0, SD = 4.1; bipolar disorder: 5.3, SD = 6.9).

General intellectual functions were assessed with the Wechsler Intelligence Scale for Children-IV (WISC-IV) (Wechsler, 2003; Bass et al., 2008). Health cards of the children were obtained from their home districts and a detailed history was obtained from the families, including the non-affected parent. The following items were taken into consideration for the general description of the sample: obstetric complications, severe childhood illness, emotional symptoms, conduct problems, and academic impairment in the school (Table 1). After a complete description of the study, all parents (including the non-affected parents) gave their written informed consent. The study was approved by the institutional ethics board and was done in accordance with the Declaration of Helsinki.

Table 1 Demographic characteristics of the participants.

	Children of mothers
with schizophrenia
(n = 28)	Children of mothers
with bipolar disorder
(n = 23)	Control children
(n = 29)	
Mean age (years)	10.6 (SD = 0.8)	10.8 (SD = 0.5)	10.6 (SD = 0.9)	
Gender (boys/girls)	20/8	18/5	16/3	
IQ	96.6 (SD = 12.4)	102.8 (SD = 11.3)	101.9 (SD = 10.5)	
Obstetric complications	1/28	1/23	1/29	
Severe childhood illness	1/28	1/23	2/29	
Emotional symptoms	6/28	4/23	0/29	
Conduct problems	3/28	4/23	1/29	
Academic impairment	5/28	2/23	1/29	

Folk psychology

The modified version of the Eyes Test was used (Baron-Cohen et al., 2001). The test consists of 28 photographs of the eye region of faces. The participant is asked to choose which of 4 words best describes what the person on the photograph is thinking and feeling. The eye regions reflect complex mental states and social emotions such as surprised, friendly, sure about something, worried or joking (Fig. 1A). As a control condition, participants were asked to judge the gender of the person using information from the eye region only.

Figure 1 Experimental stimuli.

Examples of stimuli used in the folk psychology (“theory of mind”, Eyes Test) (A) and folk physics (B) experiments (Baron-Cohen et al., 2001).

Folk physics

The folk physics test of Baron-Cohen et al. (2001) was used. The test consists of 20 items presented on separate cards. Items comprised physical problems based on the causal interactions of inanimate non-agents. Participants were asked to choose the right answer from 4 alternatives. Examples are shown in Fig. 1B. The problems presented by the folk physics test could be solved using common experiences of the physical world and were based on causal relationships. Similar to that reported by Baron-Cohen et al. (2001), our survey revealed that these problems are not included in standard school curriculum. Both folk psychology and folk physics tests have good psychometric properties.

Data analysis

The STATISTICA 6.0 (StatSoft, Inc., Tulsa) software was used for data analysis. The normality of data distribution was checked using Kolmogorov–Smirnov tests. The percentage of correct judgments in the folk psychology and folk physics tests was entered into an analysis of variance (ANOVA), followed by planned comparisons with F tests. Analyses of covariance (ANCOVAs) were applied to control the effect of IQ in relation to folk psychology and folk physics performance. The level of significance was alpha <.05.

Results

The demographic parameters of the participants are depicted in Table 1. In each group, children judged the gender of the actors and actresses in the Eyes Test with a high accuracy (>90%). A one-way ANOVA conducted on the IQ scores did not indicate a significant main effect of group (p > .1).

The ANOVA conducted on the folk psychology and folk physics performances indicated significant main effects of group (F(2, 77) = 9.8, p < .001) and task type (F(1, 77) = 97.42, p < .001). The two-way interaction was not significant (p = .6). Planned comparisons revealed that the children of mothers with schizophrenia performed worse on the Eyes Test relative to the control participants (F(1, 77) = 10.32, p < .005). In the case of the folk physics test, the difference between the children of mothers with schizophrenia and the controls reached the level of statistical significance (F(1, 77) = 3.84, p = 0.05) (Fig. 2). When the children of mothers with bipolar disorder were compared with the controls, there were no significant between-group differences for either the Eyes Test or the folk physics test (F < 1, p > .3) (Fig. 2). There was no significant difference between boys and girls (p > .1).

Figure 2 Results of the experiments.

Mean performances on the folk psychology (“theory of mind”, Eyes Test) and folk physics tests. Error bars indicate 95% confidence intervals. CONT, children with negative family history for psychotic disorders (controls); BPD, children of mothers with type I bipolar disorder; SCZ, children of mothers with schizophrenia. ∗p < .005, F test, significant after controlling for IQ (SCZ < CONT).

Next, we investigated the covariance between the folk psychology and folk physics tests and IQ. In the case of the Eyes Test, the main effect of group remained significant when IQ was included in the ANCOVA (F(2, 76) = 6.36, p < .005), and the effect of IQ was not significant (p = .3). However, in the case of the folk physics test, the main effect of the group was not significant (p = 0.1), but the effect of IQ exceeded the level of statistical significance (F(1, 76) = 10.46, p < .005).

Discussion

We found that folk psychology (“theory of mind”) was significantly impaired in children of mothers with schizophrenia but not in children of mothers with bipolar disorder. This is consistent with the view that schizophrenia vulnerability is associated with more severe early impairments than vulnerability to bipolar disorder (Murray et al., 2004). Although the children of mothers with schizophrenia also achieved lower scores on the folk physics test as compared with the controls, this difference did not reach the level of statistical significance when it was corrected for IQ. It is important to note that the folk physics test was more difficult than the folk psychology test, which is consistent with the findings of Baron-Cohen et al. (2001). Moreover, folk physics showed a significant covariance with IQ in contrast to folk psychology, which suggests that the former one is more strongly influenced by general intellectual abilities and logical reasoning. Differences among groups in folk physics therefore may be explained by differences in IQ.

The fact that we found intact performances on the gender identification test is against a generalized and profound deficit of facial processing, although the difficulty of this task was not matched to that of the folk psychology procedure. Altogether, our results suggest that offspring of individuals with schizophrenia show more pronounced impairments in the understanding of mental states of agents than in the understanding of physical causality. This may contribute to social dysfunctions (Fett et al., 2011).

Baron-Cohen et al. (2001) raised the possibility that folk psychology and folk physics are potentially dissociable, which is consistent with the present results. Children with Asperger syndrome displayed impaired performances on the Eyes Test, whereas they outperformed the healthy control group on the folk physics test. This suggests an extremely “technical brain” and a poorly functioning “social brain” in Asperger syndrome. The possible existence of extreme “social brain” and “master mindreaders” has also been suggested (Dziobek et al., 2005). Indeed, evidence from brain imaging and lesion studies indicates that several brain areas, including the dorsomedial prefrontal cortex, superior temporal cortex, and amygdala specifically participate in folk psychology and may be especially affected in psychiatric disorders such as autism-spectrum disorders and schizophrenia (Adolphs, 2003; Lee et al., 2004). This neuronal system is qualitatively different from that activated during logical reasoning (Goel & Dolan, 2001).

Fahim et al. (2004) found impaired activations of cortical areas related to emotional memory in a discordant twin pair for schizophrenia. Brain structures associated with the processing of facial information, such as the fusiform gyrus, may be especially vulnerable (Onitsuka et al., 2003; Mancini-Marie et al., 2004). Platek et al. (2005) demonstrated a relationship between abnormal medial prefrontal activation, schizotypal traits, and Eyes Test performance, which raises the possibility that subclinical symptoms may be present in children at risk of schizophrenia. Indeed, well-functioning adult relatives of schizophrenia patients without schizotypal traits show normal performances on the Eyes Test (Kelemen et al., 2004). Others found subtle deficits (Montag et al., 2012). The Eyes Test is not a simple emotion recognition paradigm because the words include both affective and non-affective mental state terms and brain areas specifically related to “theory of mind” are activated during the task (Calder et al., 2000).

The present study is not without limitations. First, the sample size was small, which results in low statistical power. However, the participants comprised a unique population (children of mothers with psychotic disorders), which limits the recruitment process. Second, there was no comprehensive neuropsychological assessment and evaluation of subclinical symptoms due to time limitations and the availability and willingness of the participants. We had no information about the potential presence of endophenotypes in the control group.

In conclusion, the present findings may facilitate the delineation of cognitive foundations of developmental abnormalities in children at potential risk of psychosis. One of the most important aims of future research is to find endophenotypes that support the identification of susceptibility and modifying genes and their interactions with environmental factors (Gottesman & Gould, 2003; Kéri & Janka, 2004), and to establish clinically useful behavioral and neuronal phenotypes that predict the development of different types of psychoses in high-risk individuals (e.g., Dazzan et al., 2012). These neurobehavioral phenotypes should be complemented by complex genetic-molecular-cellular biomarkers, taking into consideration the inherent ethical aspects of such approaches.

Supplemental Information

Supplemental Information 1 Raw data from the experiments

Click here for additional data file.

Additional Information and Declarations

Competing Interests

Author Contributions

Human Ethics

The authors declare there are no competing interests.

Rebeka Maróthi performed the experiments, analyzed the data, wrote the paper, prepared figures and/or tables, reviewed drafts of the paper.

Szabolcs Kéri conceived and designed the experiments, analyzed the data, contributed reagents/materials/analysis tools, wrote the paper, reviewed drafts of the paper.

The following information was supplied relating to ethical approvals (i.e., approving body and any reference numbers):

The Institutional Review Board of University of Szeged approved the study (2697/2010). All participants gave written informed consent.

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
