# Peer review of "Intuitive physics and intuitive psychology (“theory of mind”) in offspring of mothers with psychoses"

_PeerJ, doi:10.7717/peerj.330_

## Round 0.1 · original submission · Minor Revisions

· Academic Editor

Minor Revisions

The reviewers comments are clear and constructive, so you should have little trouble with the revision. The paper is interesting and the conclusions are well justified.

Reviewer 1 ·

Basic reporting

This manuscript appears to meet all indicated standards of reporting. It is concise, gives a sufficient background and introduction, figures are relevant. Hypotheses are clearly set out, tested, and results reported adequately.

Experimental design

The research hypotheses are clearly stated and rigorously tested. Methodology is sound.

Validity of the findings

Results are clear, conclusions are discussed in a relevant context and in a clear and concise manner.

Additional comments

This is a nice manuscript with a clear hypothesis, sound methodology, and decent sample size. Altogether, a welcome contribution to the field. Publication is to be recommended.

·

Basic reporting

The paper deals with the highly relevant questions whether or not theory of mind skills are selectively impaired in children of mothers with psychosis. To this end, the authors examined a group of children of mothers with schizophrenia, another group of offspring from mothers with bipolar disorder and a control group.
The main finding of the study was that children of mothers with schizophrenia performed selectively more poorly on a theory of mind task (but not a “folk physics” task), whereas this was not the case for children of mothers with bipolar disorder.
The paper is elegantly written and comprehensive. The introduction is sound. Methods and results are adequately presented and nicely illustrated.
The discussion refers to the relevant literature in the field.

Experimental design

Sound.

Validity of the findings

There is one confusing point in the results section (page 6, lines 126-128):

“In the case of the folk physics test, the difference between the children of mothers with schizophrenia and the controls did not reach the level of statistical significance (F(1,77) = 3.84, p = 0.05)”.

P = 0.05 should be regarded as significant. Is this a typo?

Additional comments

The paper is nicely written. The findings are interesting and worth reporting. One minor issue needs to be clarified (see above).

Signed

Martin Brüne

·

Basic reporting

The manuscript is written concisely and clearly; the message is straightforward and the proportion between the "message" and the length of the manuscript is appropriate.

However, the authors should have check the text more carefully: they systematically write "folk psychics" instead of "folk physics". This should be corrected.

Experimental design

The experimental design is straightforward: children of mothers with schizophrenia (n=28) and bipolar disorder (n=23) as well as healthy control (n=29) were engaged to understand mental states of others using the Eyes Test (folk psychology or “theory of mind”) and physical causal interactions of inanimate objects (folk physics).

What is not clear at all is the following: Apart from the general categorisation of being in the same disease category according to DSM-IV, to what extent were the two groups of the diseased mother (schizophrenia and bipolar) mothers homogenous (age of onset, number of acute episodes, time of the disease, severity, etc.)?

Validity of the findings

The conclusions are clearly convincing (and basically confirm our basic expectations and experience with such children), but the authors should definitely call the readers' attention to the limitations of their study regarding the sample size (and, consequently, the statistical power of the observations), as well as to other confounders (possible inhomogeneity of the groups of mothers; were among the mothers of the "control group" some endophenotypes?, etc.).

Additional comments

It would be important to indicate in the discussion the necessity of having a more detailed genomic-phenomic-metabonomic (even, biota!) type information of the subjects (both mothers and children) and, consequently, a detailed analysis of correlation between the outcome measure and the underlying genetic-molecular-cellular biomarkers.

---

## Round 0.2 · accepted · Accept

· Academic Editor

Accept

Your article is an interesting and valuable contribution to our understanding of susceptibility to schizophrenia.